# Triaging HPV-Positive Cervical Samples with p16 and Ki-67 Dual Stained Cytology within an Organized Screening Program—A Prospective Observational Study from Western Norway

**DOI:** 10.3390/ijms24087158

**Published:** 2023-04-12

**Authors:** Irene Tveiterås Øvestad, Ingvild Dalen, Marie S. Andersland, Olav K. Vintermyr, Pia Moltu, Jannicke M. Berland, Emilius A. M. Janssen, Hans Kristian Haugland

**Affiliations:** 1Department of Pathology, Stavanger University Hospital, 4011 Stavanger, Norway; 2Department of Research, Stavanger University Hospital, 4011 Stavanger, Norway; 3Department of Pathology, Haukeland University Hospital, 5053 Bergen, Norway; 4Gade Laboratory for Pathology, Department of Clinical Medicine, University of Bergen, 5020 Bergen, Norway; 5Department of Chemistry, Bioscience and Environmental Technology, University of Stavanger, 4021 Stavanger, Norway

**Keywords:** HPV, cytology, p16, Ki-67, immunocytochemistry, organized screening program

## Abstract

The implementation of high-risk human papillomavirus testing (hrHPV testing) as a screening method in substitute for cytology has evoked the need for more sensitive and less objective tests for the triage of HPV-positive women. In a cohort of 1763 HPV-positive women, the potential of immunocytochemical p16 and Ki-67 dual staining as compared to cytology, alone or in combination with HPV partial genotyping, was tested for triage of women attending a cervical cancer screening program. Performance was measured using sensitivity, specificity, and positive and negative predictive values. Comparisons were assessed using logistic regression models and the McNemar test. Dual staining was evaluated in a prospectively collected study cohort of 1763 HPV-screened women. For triage of CIN2+ and CIN3+, NPV and sensitivity, 91.8% and 94.2% versus 87.9% and 89.7%, respectively, were significantly higher using dual staining together with HPV 16/18 positive, as compared to cytology (*p* < 0.001). The specificities, however, were lower for dual staining as compared to cytology. Conclusions: Dual staining is safer for decision-making regarding HPV-positive women’s need for follow-up with colposcopy and biopsy, as compared to cytology.

## 1. Introduction

Persistent infection with high-risk human papillomavirus (hrHPV) is, with rare exceptions, a prerequisite for developing cervical intraepithelial neoplasia (CIN) and invasive carcinoma [1]. During a lifetime, the vast majority of sexually active women are exposed to a genital HPV infection [2], but more than 90% of infections clear within 1–2 years without doing any harm [3,4].

In developed countries, organized cervical cancer screening programs (CCSP) have existed for decades, and despite increasing exposure to hrHPV in the population, the incidence of cervical cancer is kept at a low level compared to countries without such programs [5]. For quite some time, HPV testing has been used as a triage of atypical squamous cells of undetermined significance (ASC-US) and low-grade squamous intraepithelial lesions (LSIL). In later years, based on results from international studies [6,7,8,9], many countries with organized screening programs have replaced cytology, as the primary screening method, with an hrHPV test for women 30 years and older. HPV screening is more sensitive than cytology screening but less specific, and a negative HPV test is a long-time predictor for not developing cervical cancer [10,11]. In order to increase the accuracy and positive predictive value; secondary cytology screening is performed on HPV-positive samples. Adding information regarding the HPV genotypes further reduces over-referral to colposcopy and increases risk stratification [12]. A pilot implementation of HPV screening in four Norwegian counties showed a significant increase in the number of diagnosed CIN lesions. For women randomized to HPV screening as compared to liquid-based cytology (LBC) screened women, there was a 60% increase in colposcopies and 40% more normal, and 2.7 times more CIN1 biopsies were diagnosed [13]. With delayed follow-up if negative for intraepithelial lesion or malignancy (NILM) at 12 months when positive for HPV 16/18, NILM at 24 months, and ASC-US/LSIL at 12 months when positive for the 12 other hrHPV, 66% of the women would be referred to colposcopy, with a negligible change for detection of CIN3+ [14]. It is a concern that implementing a more sensitive but less specific screening method will increase the risk of over-referral and overdiagnosis with colposcopy and biopsy and, eventually, over-treatment [15,16,17]. Since many countries have implemented HPV screening for women > 30 years of age or are planning to do so, there is a growing need for more accurate methods to supplement follow-up decisions of HPV-positive women.

The first cohorts of vaccinated women protected against the most carcinogenic HPV subtypes, 16 and 18, have entered cervical cancer screening programs, and the number will increase in the years to come. HPV 16 and 18 account for 70% of all cervical cancers worldwide. However, the vaccinated population will still be vulnerable to infection with 12 other high-risk (hr) HPV subtypes (31, 33, 35, 39, 45, 51, 52, 56, 58, 59, 66, and 68). The oncogenic potential of these hrHPV subtypes is very heterogenic, with a variable prevalence in cervical cancers ranging from 1 to 6% [18], and it implies an increasing chance for over-diagnosis and over-treatment.

Immunocytochemical p16/Ki-67 dual staining (dual staining) for detection of co-expression of p16 and Ki-67 in the same cell has been suggested as a suitable method to increase the specificity and sensitivity for triage of HPV-positive women [19,20,21,22]. p16 (also known as p16INK4a) is a cyclin-dependent kinase inhibitor and becomes overexpressed when the cell-cycle regulatory retinoblastoma protein (pRb) is inactivated by the high-risk HPV E7 oncoprotein. p16 expression is thus a surrogate marker for this event [23], while Ki-67, a marker of cell proliferation and a key feature of CIN progression, is highly correlated with the grade of dysplastic change in the epithelium [24]. The current prospective observational study tested the ability of dual staining for triage of HPV-positive women, alone or in combination with partial HPV genotyping, defined as HPV 16 or 18 positive or positive for the group of 12 other hrHPV, as compared to the current triage with cytology testing and partial HPV genotyping.

## 2. Results

### 2.1. Biological Material

The mean age of the study cohort was 46 years (range, 33–69). As shown in Table 1, 65% of LBC samples used for the dual staining (Index cytology) were normal, and 35% were ASC-US or worse. In total, 48% of the index cytology were dual stained positive, 74% were positive for 12 other HPV, and 26% were positive for either HPV 16 or 18. After a median of 370 days (IQR, 50–571), 981 women had a valid dual staining result and follow-up with histology, punch biopsy (*n* = 850), or cone biopsy (*n* = 131), diagnosed as normal (34%), CIN1 (25%), CIN2 (7%), CIN3 (31%), Adenocarcinoma in situ (AIS) (1%) or cervical carcinoma (CC) (2%). In case of any uncertainty regarding histologic grading, immunohistochemistry for Ki-67 and p16 supported the decision-making for 70% of the biopsies.

After the median 391 days (IQR, 352–735), 434 women had an HPV-negative cytology result, defined as HPV negative, regardless of cytology diagnosis. After a median of 368 days (IQR, 332–828 days), 63 women who had a follow-up with non-conclusive positive cytology, defined as NILM, ASCUS or LSIL, and HPV positive and were excluded from the study. Details regarding study samples are shown in Figure 1 and Table 1.

### 2.2. p16/Ki-67 Dual Staining Results

The dual stain analysis was performed median of 120 (IQR, 84–158) days after sampling. However, regression models showed no statistically significant time dependency for the outcome of the test (*p* = 0.29). In Table 2, the sensitivity, specificity, PPV, and NPV, including two-sided 95% CI, are listed according to triage strategies and clinical endpoints, as shown in Box 1. The percentage of dual stained positives increased with the severity of the biopsy diagnosis, Normal (35%), CIN1 (49%), CIN2 (74%), CIN3 (83%), AIS (92%), and CC (94%), as shown in Table 1.

Box 1Definitions of triage strategies and outcomes. Abbreviations: ASC-US: Atypical Squamous cells—Undetermined Significance, NILM: Negative for Intraepithelial Lesions or Malignancy.
**Triage Strategies**

Dual stainingPositive: At least one observed p16/Ki-67 dual stained epithelial cell Negative: No p16/Ki-67 dual stained epithelial cells observed CytologyPositive: ASC-US or worse cytology Negative: NILM cytologyHPV 16/18Positive: HPV genotype 16 and/or 18 present, possibly also 12 other high risk genotypesNegative: No HPV genotype16/18 present i.e., only 12 other high risk genotypesDual staining and HPV 16/18Positive: Positive dual staining **and** positive HPV 16/18 Negative: Negative dual staining **and/or** negative HPV 16/18Cytology and HPV 16/18Positive: Positive cytology **and** positive HPV 16/18 Negative: Negative cytology **and/or** negative HPV 16/18Dual staining and/or HPV 16/18Positive: Positive dual staining **and/or** positive HPV 16/18 Negative: Negative dual staining **and** negative HPV 16/18Cytology and/or HPV 16/18Positive: Positive cytology **and/or** positive HPV 16/18 Negative: Negative cytology **and** negative HPV 16/18
**Outcomes**

CIN2+Positive: CIN2 or worse histology; Negative: CIN1 or less histology or no histology but negative follow-up HPV testCIN3+Positive: CIN3 or worse histology; Negative: CIN1 or less histology or no histology but negative follow-up HPV test

The sensitivities for detecting CIN2+ or CIN3+ for triage strategies using dual staining, either alone or in combination with HPV16/18, were significantly higher than for cytology alone or in combination with HPV16/18, but the specificities for triage strategies involving cytology were significantly higher than for those involving dual staining, Table 2.

The PPVs for CIN2+ and CIN3+ given a positive triage using dual staining alone or in combination with HPV16/18 were generally lower within this sample than the corresponding PPVs for cytology, but none of these differences were statistically significant. The NPVs were, on the other hand, statistically significantly higher for triage strategies involving dual staining than for cytology. The highest NPV observed was 91.8% (95% CI 89.4 to 93.8%) for CIN2+ and 94.2% (92.0 to 95.8%) for CIN3+. Both were obtained with triage, using dual staining and/or HPV 16/18 positive and were significantly higher (*p* < 0.001) than for the corresponding triage with positive cytology and/or positive HPV 16/18; NPVs 88.1% and 91.0% for CIN2+ and CIN3+, respectively, Box 1 and Table 2.

For dual staining, logistic regression revealed statistically significant differences between the group of HPV16/18 positive and the group of 12 other hrHPV positive in the sensitivity to detect CIN2+ (*p* = 0.001) and the sensitivity (*p* = 0.017) and specificity (*p* = 0.003) with regard to detection of CIN3+. For cytology, the differences in sensitivity and specificity between the HPV groups were not statistically significant, Table 3. Differences regarding triage, however, remained unchanged between the two methods.

After excluding women with diagnoses that were established more than two years after the index cytology (*n* = 304; 21%), the results remained similar to the presented main findings (see Appendix A).

### 2.3. Expected Follow-Up Results by Use of p16/Ki-67 Dual Staining versus Cytology

Based on the results presented in Table 2, with triage requiring both positive cytology and HPV 16/18 genotype, 167 biopsies would be performed, with about 43 being negative for CIN2+. On the other hand, of the 1248 women not biopsied, 280 would be CIN2+.

With a similar triage based on dual staining instead of cytology, 246 biopsies would be performed, with 73 being negative, i.e., almost 70% more than with cytology. On the other hand, 1169 women would not be biopsied, and of these, 231 would be CIN2+, i.e., 17% less than with cytology.

Corresponding results for a triage where those who have either positive cytology or HPV 16/18 genotype (or both) are biopsied, 402 (of 724) biopsies would be negative, and 82 CIN2+ would be missed. With dual staining replacing cytology, 459 (of 814) biopsies would be negative (+14% vs. cytology), and 49 CIN2+ would be missed (−40% vs. cytology).

## 3. Discussion

Demands for a triage test to distinguish between persistent infections with the ability to develop high-grade pre-cancer and harmless transient infections are urgent in order to decrease the costs for over-screening and the burden on colposcopy clinics, as a result of introducing the less specific HPV primary screening for women ≥ 30 years [13,16,25].

The purpose of the current prospective multicenter study was to evaluate the utility of dual staining as compared to cytology in the triage of HPV-positive cervical samples within an organized CCSP. For predicting cervical pre-cancer, CIN2+, and CIN3+, screening with dual staining, alone or in combination screening with partial genotyping, demonstrated higher sensitivities and higher NPVs, but lower specificities as compared to cytology screening, while the PPV was quite similar for the two, (Table 2).

Comparing cumulative results and HPV detection by Cobas 4800, the results in the current study are similar to the results found by Wright et al. [22], showing higher sensitivities but lower specificities for dual staining as compared to cytology, either alone or in combination with HPV16/18 genotyping. The current study, however, demonstrated much higher specificities, especially for dual staining and cytology in combination with HPV16/18 genotyping, while the sensitivities and NPV were somewhat lower. This was also the case when comparing results from the two studies using triage with dual staining and cytology in stratified groups of HPV16/18 positives and 12 other hrHPV positives. One explanation for the lower specificities could be women with a younger age group at enrolment in the study by Wright et al., 25–65 versus 34–69 years, as younger women tend to have more frequent and transient HPV infections, cleared within 1–2 years without doing any harm, as stated by Munoz et al. [26]. As compared to our results, the sensitivities and NPV for triage with both cytology and dual staining in combination with HPV partial genotyping are much in line with the study by Wentzensen et al. [27], while also their specificities were considerably lower. The current study found 58% of dual stained positives among women referred for colposcopy and comparable to the study by McMenamin et al., who found 59.7% dual stained positives in the same age group [28].

Based on the calculated sensitivities and specificities for triage requiring both positive dual staining and HPV 16/18 would entail nearly 70% more negative biopsies than a similar triage using positive cytology. Additionally, 231 women with CIN2+ would be excluded from taking a biopsy, 17% less than with positive cytology and HPV 16/18.

Using a strategy with either positive dual staining or positive for HPV 16/18 as triage would mean 40% less missed CIN2+ and a 14% increase in negative biopsies as compared to using the same strategy with positive cytology. The group with positive dual staining and not positive for HPV 16/18 comprise the 12 other hrHPV positives, the largest group in the current cohort and a growing concern as women vaccinated for the most oncogenic HPV 16/18 enter screening programs. The use of a more sensitive method in screening programs would probably also lead to higher numbers of detected CIN2 lesions with a high regression rate. Many pathologists perceive CIN2 as an equivocal diagnosis, and a meta-analysis has concluded that conservative management with active surveillance rather than immediate ablation of CIN2 was a safe alternative, especially for young women [29]. Dual staining with high sensitivity and high NPV would also be advantageous as a less invasive follow-up strategy for women with CIN2 lesions and a high possibility for regression.

The present study has its strengths and weaknesses. The use of follow-up data from the laboratory information system at the two pathology departments gave an unbiased description of the decision-making the pathologists and cytotechnologists are faced with in daily practice. More than two years of follow-up before the final diagnosis for 20% of the study population indeed, had an impact on the sensitivities and specificities of both methods to detect CIN2+ and CIN3+. On the other hand, comparisons of the two methods in the whole population and in the group diagnosed within 2 years did not show conspicuous differences. The long follow-up was due to an updated version of the algorithm for follow-up testing of HPV-positive women introduced in the screening program during the study period. The new version recommends delayed follow-up in 24 months for women with NILM cytology and positive for 12 other hrHPV. The long duration of follow-up for this group of women might have increased the chance of dropping out of testing and contributed to longer follow-up before a final diagnosis. The explanation for the majority of 9% invalid p16/Ki-67 dual staining results was low cellularity and expected, as the smear for dual staining was reprocessed after the smear for cytology screening. Additionally, samples contaminated with blood and treated with Glacial acetic acid (GAA) have been observed to entail brown artifacts, which may affect the validation of the staining results [30]. This is a commonly used method if smears are intended for use in cytology screening, but it should not be necessary if smears are used for screening with dual staining.

In conclusion, based on our findings, positive dual staining and/or positive HPV 16/18 versus negative dual staining and/or negative HPV 16/18 is a safe alternative for the triage of HPV-positive women from an HPV-screened population. Replacing the current cytology screening with dual staining and/or positive HPV 16/18 would result in a wanted outcome with the detection of an increased number of women with CIN2+, but at the expense of an increased number of proven negative biopsies. As compared to the present triage with cytology in combination with partial genotyping, this strategy demonstrated significantly higher sensitivities and higher NPVs. Consequently, dual staining would provide safer decision-making for whether HPV-positive women need follow-up with colposcopy and biopsy or if watchful waiting with cytology-based follow-up is more beneficial.

## 4. Materials and Methods

### 4.1. Biological Material

From October 2017 until November 2018, residual material from liquid-based samples fixed in Preservcyt (Hologic, Inc., Marlborough, MA, USA) were prospectively collected from 1763 HPV-positive women who had been triaged with cytology. All women were assigned to primary HPV screening in a pilot study, including two counties in the western part of Norway [13]. Excluded were women that had a known history of cervical therapy within the past 24 months. In total, 1164 samples were collected at the Department of Pathology, Haukeland University Hospital (HUH), and 599 samples at the Department of Pathology, Stavanger University Hospital (SUH). Details regarding study samples are shown in Figure 1. Follow-up diagnoses were retrieved from the pathology patient registry system at SUH and HUH between January 2018 and December 2021. The endpoints were defined as either an HPV-negative cytology test or the worst detected histology diagnoses during follow-up time, as described in BOX 1. If multiple results were registered, the most severe histology diagnosis was used for analysis. For 70% of the biopsies, immunohistochemical markers for Ki-67, p16, or Ki-67 and p16 were used to support the grading of CIN.

The study was approved by the Regional Ethical Committee (REK Sør-øst), Norway, 2017/748. Since the study participants followed the algorithm of an organized CCSP and no intervention took place based on the study result; informed consent was waived.

### 4.2. HPV Screening and Cytology Triage

Primary HPV screening was done on the Cobas 4800 system (Roche Molecular Diagnostics, Pleasanton, CA, USA), a qualitative HPV DNA test for the detection of 14 high-risk HPV (hrHPV) in cervical specimens. Cobas 4800 specifically identifies high-risk HPV 16 and 18 and detects the 12 other hrHPV (12 other HPV) as a group. The testing was performed according to the manufacturer’s recommendations, and the protocol provided by Roche was followed.

All HPV-positive samples were triaged with liquid-based cytology processed on a ThinPrep 5000 processor (Hologic, Inc., Marlborough, MA, USA), screened by experienced cytotechnicians, according to the Bethesda System [31] and signed out by trained cytopathologists at SUH and HUH. The diagnostics were performed in a routine setting, and accordingly, cytotechnicians and cytopathologists were not blinded to the HPV results.

### 4.3. p16/Ki-67 Dual Staining Analysis

Residual material from LBC samples from 1763 HPV-positive women were used for dual staining, performed at Haukeland University Hospital (Bergen, Norway) by use of the CINtec^®^ PLUS Cytology kit (Roche Diagnostics GmbH, Mannheim, Germany). Smears were processed on a ThinPrep 5000 processor, and immunocytochemistry was performed on a fully automated slide-stainer platform (Ventana Benchmark Ultra; Roche Diagnostics) as described in the package insert. Laboratory technicians, who were responsible for the immunocytochemistry procedure, accomplished a 2-day technical training course from Roche. Additionally, cytotechnicians and cytopathologists taking part in the study accomplished a 2-day-course on slide interpretation and a proficiency test under the auspices of Roche.

### 4.4. Interpretation of p16/Ki-67 Dual Staining Results

The definition of a p16/Ki-67 dual-stained cell is a cervical epithelial cell with simultaneous brown cytoplasmic immunostaining and red immunostaining of the nucleus. Samples were considered dual stained positive if at least one cervical epithelial cell was stained with both a red nuclear stain for Ki-67 and a brown cytoplasmic stain for p16. The screening was performed by cytotechnicians, and findings were confirmed by a cytopathologist, both blinded to the result of the triage cytology. Microscopy was done at a screening magnification of 10× while evaluation of double-stained cells was done at 40×. If no double-stained cells were found, the staining result was considered negative. One hundred and sixty samples (9%) were scored as inconclusive, (i) if the smears had less than four cells per field of vision in a minimum of 10 fields of vision with a 40× objective, (ii) if not at least one cell on the slide contained one red signal (Ki-67) and one brown signal (p16) or (iii) if the sample was contaminated by blood. For 37 (2%) cases, the discrepancy was observed between the cytotechnician and cytopathologist regarding validation of the dual staining result, and a consensus was achieved in a plenary session using a multiheaded microscope.

### 4.5. Statistical Methods

Descriptive statistics are presented as counts and percentages (%) for categorical data and as medians and interquartile ranges (IQR) for continuous data. The association between time from index cytology sampling to the p16/Ki-67 dual staining analysis and the odds of a positive test was assessed in logistic regression models, with and without allowing for non-linear association using restricted cubic splines. Sensitivities (SE), specificities (SP), positive (PPV), and negative predictive values (NPV) were estimated and presented with two-sided 95% Wilson confidence intervals (CI). SEs and SPs were compared between different triage strategies within the same sample using the McNemar test. PPVs and NPVs were compared between triage strategies within the same sample using the approach suggested by Leisenring [32]. Furthermore, comparisons between SEs and SPs obtained in different samples were performed using logistic regression. AS-CUS (Atypical Squamous cells of Undetermined Significance) or worse versus NILM (Negative for Intraepithelial Lesions or Malignancy) cytology and dual stained positives versus dual stained negatives in the whole cohort or in combination with partial genotyping, as listed in Box 1. The statistical analyses were performed in Stata v. 17.0 with functions logit, cii, mcc, and xtgee. *p* values < 0.05 were considered statistically significant.

## Figures and Tables

**Figure 1 ijms-24-07158-f001:**
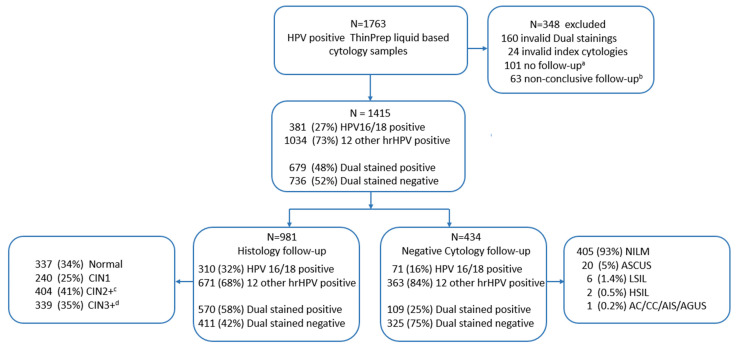
Flow Scheme of the study cohort showing partial HPV genotyping and dual staining results of the index cytology of samples included in the study. ^a^ missing follow-up after index cytology; ^b^ follow-up with HPV positive cytology, regardless of cytology diagnosis. ^c^ CIN2, CIN3, AIS/CIS and CC; ^d^ CIN3, AIS/CIS and CC.

**Table 1 ijms-24-07158-t001:** Cervical cytology results, partial HPV genotyping, and p16/Ki-67 dual staining results in baseline cytology and follow-up with either HPV negative or positive cytology or normal, CIN1, CIN2, CIN3, AIS or CC histology from an HPV screened population. (*n* = 1478).

Index Cytology, HPV Genotypes and Dual Staining Results	Total	^a^ HPV Neg. Follow-Up Cytology	^a^ HPV Pos. Follow-Up Cytology	Normal	CIN1	CIN2	CIN3	AIS/ CIS	CC
	**(*N* = 1478)**	**(*N* = 434)**	**(*N* = 63)**	**(*N* = 337)**	**(*N* = 240)**	**(*N*= 65)**	**(*N* = 309)**	**(*N* = 13)**	**(*N* = 17)**
HPV 16 or 18 HPV 12 other	**390 (26%)**	**71 (16%)**	**9 (10%)**	**63 (19%)**	**53 (22%)**	**27 (42%)**	**145 (47%)**	**10 (77%)**	**12 (71%)**
	**1088 (74%)**	**363 (84%)**	**54 (90%)**	**274 (81%)**	**187 (78%)**	**38 (58%)**	**164 (53%)**	**3 (23%)**	**5 (29%)**
Dual stained pos.	**703 (48%)**	**109 (25%)**	**24 (38%)**	**117 (35%)**	**119 (49%)**	**48 (74%)**	**258 (83%)**	**12 (92%)**	**16 (94%)**
Dual stained neg.	**775 (52%)**	**325 (75%)**	**39 (62%)**	**220 (65%)**	**121 (51%)**	**17 (26%)**	**51 (17%)**	**1 (8%)**	**1 (6%)**
**NILM**	**961 (65%)**	**405 (93%)**	**55 (87%)**	**217 (64%)**	**132 (55%)**	**33 (51%)**	**108 (34%)**	**3 (23%)**	**8 (47%)**
HPV 16 or 18	220 (23%)	69 (17%)	9 (16%)	43 (20%)	32 (24%)	13 (39%)	49 (45%)	2 (67%)	6 (75%)
HPV 12 other	741 (77%)	336 (83%)	46 (84%)	174 (80%)	100 (76%)	20 (61%)	59 (55%)	1 (33%)	2 (25%)
Dual stained pos.	324 (34%)	92 (23%)	20 (36%)	59 (27%)	48 (37%)	20 (61%)	75 (69%)	3 (100%)	7 (88%)
Dual stained neg.	637 (66%)	313 (77%)	35 (64%)	158 (73%)	84 (63%)	13 (39%)	33 (31%)	-	1 (12%)
**ASC-US**	**222 (15%)**	**20 (5%)**	**7 (11%)**	**70 (21%)**	**61 (25%)**	**14 (22%)**	**47 (15%)**	**2 (15%)**	**1 (6%)**
HPV 16 or 18	58 (26%)	1 (5%)	-	11 (16%)	15 (25%)	5 (36%)	23 (49%)	2 (100%)	1 (100%)
HPV 12 other	164 (74%)	19 (95%)	7 (100%)	59 (84%)	46 (75%)	9 (64%)	24 (51%)	-	-
Dual stained pos.	128 (58%)	9 (45%)	3 (43%)	28 (40%)	35 (57%)	10 (71%)	41 (87%)	1 (50%)	1 (100%)
Dual stained neg.	94 (42%)	11 (55%)	4 (57%)	42 (60%)	26 (43%)	4 (29%)	6 (13%)	1 (50%)	-
**LSIL**	**117 (8%)**	**6 (2%)**	**1 (2%)**	**30 (9%)**	**38 (16%)**	**9 (14%)**	**33 (11%)**		
HPV 16 or 18	30 (26%)	-	-	6 (20%)	5 (13%)	5 (56%)	14 (42%)	-	-
HPV 12 other	87 (74%)	6 (100%)	1 (100%)	24 (80%)	33 (87%)	4 (44%)	19 (58%)	-	-
Dual stained pos.	92 (79%)	5 (86%)	1 (100%)	17 (57%)	31 (82%)	9 (100%)	29 (88%)	-	-
Dual stained neg.	25 (21%)	1 (14%)	-	13 (43%)	7 (18%)	-	4 (12%)	-	-
**ASC-H**	63 (4%)	-	-	11 (3%)	6 (2%)	3 (5%)	38 (12%)	3 (23%)	2 (12%)
HPV 16 or 18	22 (34%)	-	-	2 (18%)	-	2 (67%)	15 (40%)	2 (67%)	1 (50%)
HPV 12 other	41 (66%)	-	-	9 (82%)	6 (100%)	1 (33%)	23 (60%)	1 (33%)	1 (50%)
Dual stained pos.	51 (80%)	-	-	6 (55%)	3 (50%)	3 (100%)	34 (90%)	3 (100%)	2 (100%)
Dual stained neg.	12 (20%)	-	-	5 (45%)	3 (50%)	-	4 (10%)	-	-
**HSIL**	107 (7%)	2 (0.5%)	-	9 (3%)	3 (1%)	6 (9%)	80 (26%)	3 (23%)	4 (22%)
HPV 16 or 18	52 (48%)	1 (50%)	-	1 (11%)	1 (33%)	2 (33%)	43 (54%)	2 (67%)	2 (50%)
HPV 12 other	55 (52%)	1 (50%)	-	8 (89%)	2 (67%)	4 (67%)	37 (46%)	1 (33%)	2 (50%)
Dual stained pos.	100 (94%)	2 (100%)	-	7 (70%)	2 (67%)	6 (100%)	76 (95%)	3 (100%)	4 (100%)
Dual stained neg.	7 (6%)	-	-	2 (30%)	1 (33%)	-	4 (5%)	-	-
**AC/CC/AIS/AGUS**	8 (0.5%)	1 (0.2%)		-	-	-	3 (1%)	2 (15%)	2 (12%)
HPV 16 or 18	5 (63%)	-	-	-	-	-	1 (33%)	2 (100%)	2 (100%)
HPV 12 other	3 (37%)	1 (100%)	-	-	-	-	2 (67%)	-	-
Dual stained pos.	8 (100%)	1 (100%)	-	-	-	-	3 (100%)	2 (100%)	2 (100%)
Dual stained neg.	-	-	-	-	-	-	-	-	-

^a^ No Follow-up biopsy Abbreviations: NILM (Negative for Intraepithelial Lesion Malignancy, ASC-US (Atypical squamous cells of undetermined significance), LSIL (Low-grade squamous intraepithelial lesion), ASC-H (Atypical Squamous Cells, Cannot Rule Out High Grade Squamous Intra-epithelial Lesion), HSIL (High-grade squamous intraepithelial lesion), AGUS (atypical glandular cells of undetermined significance), CIN1,2,3 (Cervical Intraepitelial Neoplasia grade 1,2 or 3), AIS/CIS (Adenocarcinoma/Carcinoma in situ), AC, Adenocarcinoma, CC (Cervical cancer), pos. (positive), neg. (negative).

**Table 2 ijms-24-07158-t002:** Performance of p16/Ki-67 dual staining as compared to cytology for triage of CIN2+ and CIN3+, alone or in combination with partial HPV genotyping in a cohort of primary high-risk HPV positive women (*n* = 1415).

Outcome	Triage Strategy ^1^	SE (95% CI)	SP (95% CI)	PPV (95% CI)	NPV (95% CI)	*p* Values ^2^
CIN2+	Dual staining	82.7 (78.7, 86.1) *n* = 404	65.9 (62.9, 68.7) *n* = 1011	49.2 (45.4, 52.9) *n* = 679	90.5 (88.2, 92.4) *n* = 736	<0.001 (SE) <0.001 (SP) 0.90 (PPV) <0.001 (NPV)
	Cytology	62.4 (57.6, 67.0) *n* = 404	74.5 (71.7, 77.1) *n* = 1011	49.4 (45.1, 53.7) *n* = 510	83.2 (80.6, 85.5) *n* = 905	
	HPV 16/18	48.0 (43.2, 52.9) *n* = 404	81.5 (79.0, 83.8) *n* = 1011	50.9 (45.9, 55.9) *n* = 381	79.7 (77.1, 82.0) *n* = 1034	<0.001 (SE) <0.001 (SP) 0.18 (PPV) <0.001 (NPV)
	Dual staining and HPV 16/18	42.8 (38.1, 47.7) *n* = 404	92.8 (91.0, 94.2) *n* = 1011	70.3 (64.3, 75.7) *n* = 246	80.2 (77.9, 82.4) *n* = 1169	
	Cytology and HPV 16/18	30.7 (26.4, 35.4) *n* = 404	95.7 (94.3, 96.8) *n* = 1011	74.3 (67.1, 80.3) *n* = 167	77.6 (75.2, 79.8) *n* = 1248	
	Dual staining and/or HPV 16/18	87.9 (84.3, 90.7) *n* = 404	54.6 (51.5, 57.6) *n* = 1011	43.6 (40.2, 47.0) *n* = 814	91.8 (89.4, 93.8) *n* = 601	<0.001 (SE) <0.001 (SP) 0.42 (PPV) 0.001 (NPV)
	Cytology and/or HPV 16/18	79.7 (75.5, 83.3) *n* = 404	60.2 (57.2, 63.2) *n* = 1011	44.5 (40.9, 48.1) *n* = 724	88.1 (85.5,90.3) *n* = 691	
CIN3+	Dual staining	84.4 (80.1, 87.8) *n* = 339	63.5 (60.6, 66.3) *n* = 1076	42.1 (38.5, 45.9) *n* = 679	92.8 (90.7, 94.5) *n* = 736	<0.001 (SE) <0.001 (SP) 0.54 (PPV) <0.001 (NPV)
	Cytology	64.9 (59.7, 69.8) *n* = 339	73.0 (70.3, 75.6) *n* = 1076	43.1 (38.9, 47.5) *n* = 510	86.9 (84.5, 88.9) *n* = 905	
	HPV 16/18	49.3 (44.0, 54.6) *n* = 339	80.1 (77.6, 82.4) *n* = 1076	43.8 (38.9, 48.9) *n* = 381	83.4 (81.0, 85.5) *n* = 1034	
	Dual staining and HPV 16/18	44.0 (38.8, 49.3) *n* = 339	91.0 (89.1, 92.6) *n* = 1076	60.6 (54.3, 66.5) *n* = 246	83.7 (81.5, 85.8) *n* = 1169	<0.001 (SE) <0.001 (SP) 0.065 (PPV) <0.001 (NPV)
	Cytology and HPV 16/18	32.4 (27.7, 37.6) *n* = 339	94.7 (93.2, 95.9) *n* = 1076	65.9 (58.4, 72.6) *n* = 167	81.7 (79.4, 83.7) *n* = 1248	
	Dual staining and/or HPV 16/18	89.7 (86.0, 92.5) *n* = 339	52.6 (49.6, 55.6) *n* = 1076	37.2 (34.1, 40.7) *n* = 814	94.2 (92.0, 95.8) *n* = 601	<0.001 (SE) <0.001 (SP) 0.35 (PPV) 0.002 (NPV)
	Cytology and/or HPV 16/18	81.7 (77.2, 85.5) *n* = 339	58.5 (55.5, 61.4) *n* = 1076	38.3 (34.8, 41.9) *n* = 724	91.0 (88.7, 92.9) *n* = 691	

^1^ Triage strategies defined in Box 1, ^2^ *p* values for comparison of triage involving cytology vs. triage involving dual staining. Abbreviations: SE (Sensitivity), SP (Specificity), PPV (Positive predictive value), NPV (Negative predictive value).

**Table 3 ijms-24-07158-t003:** Performance of p16/Ki-67 dual staining as compared to cytology for triage of CIN2+ and CIN3+ in a cohort of primary high-risk HPV positive women, grouped according to partial HPV genotyping (*n* = 381 HPV 16/18, *n* = 1034 12 other high-risk HPV).

Outcome	HPV Status	Triage Strategy ^1^	SE (95% CI)	SP (95% CI)	PPV (95% CI)	NPV (95% CI)	*p* Values ^2^
CIN2+	HPV 16/18	Dual staining	89.2 (84.0, 92.8) (*n* = 194)	61.0 (53.8, 67.7) (*n* = 187)	70.3 (64.3, 75.7) (*n* = 246)	84.4 (77.4, 89.6) (*n* = 135)	<0.001 (SE) <0.001 (SP) 0.18 (PPV) <0.001 (NPV)
		Cytology	63.9 (56.9, 70.3) (*n* = 194)	77.0 (70.5, 82.5) (*n* = 187)	74.3 (67.1, 80.3) (*n* = 167)	67.3 (60.7, 73.2) (*n* = 214)	
	HPV 12 other	Dual staining	76.7 (70.5, 81.9) (*n* = 210)	67.0 (63.7, 70.1) (*n* = 824)	37.2 (32.8, 41.8) (*n* = 433)	91.8 (89.4, 93.8) (*n* = 601)	<0.001 (SE) <0.001 (SP) 0.95 (PPV) 0.001 (NPV)
		Cytology	61.0 (54.2, 67.3) (*n* = 210)	73.9 (70.8, 76.8) (*n* = 824)	37.3 (32.4, 42.6) (*n* = 343)	88.1 (85.5, 90.3) (*n* = 691)	
CIN3+	HPV 16/18	Dual staining	89.2 (83.6, 93.1) (*n* = 167)	54.7 (48.0, 61.2) (*n* = 214)	60.6 (54.3, 66.5) (*n* = 246)	86.7 (79.9, 91.4) (*n* = 135)	<0.001 (SE) <0.001 (SP) 0.065 (PPV) <0.001 (NPV)
		Cytology	65.9 (58.4, 72.6) (*n* = 167)	73.4 (67.1, 78.8) (*n* = 214)	65.9 (58.4, 72.6) (*n* = 167)	73.4 (67.1, 78.8) (*n* = 214)	
	HPV 12 other	Dual staining	79.7 (73.0, 85.0) (*n* = 172)	65.7 (62.4, 68.8) (*n* = 862)	31.6 (27.4, 36.2) (*n* = 433)	94.2 (92.0, 95.8) (*n* = 601)	<0.001 (SE) <0.001 (SP) 0.82 (PPV) 0.002 (NPV)
		Cytology	64.0 (56.5, 70.8) (*n* = 172	73.0 (69.9, 75.8) (*n* = 862)	32.1 (27.4, 37.2) (*n* = 343)	91.0 (88.7, 92.9) (*n* = 691)	

^1^ Triage strategies defined in Box 1, ^2^ *p* values for comparison of triage with cytology vs. dual staining. Abbreviations: SE (Sensitivity), SP (Specificity), PPV (Positive predictive value), NPV (Negative predictive value).

## Data Availability

Data is unavailable due to privacy or ethical restrictions.

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
