# Peer review of "Triaging HPV-Positive Cervical Samples with p16 and Ki-67 Dual Stained Cytology within an Organized Screening Program—A Prospective Observational Study from Western Norway"

_ijms, 2023, doi:10.3390/ijms24087158_

Round 1

Reviewer 1 Report

In their paper entitled “Triaging HPV positive cervical samples with p16 and Ki-67 2 dual stained cytology within an organized screening programme, a prospective observational study from Western Norway”, Øvestad et al, tested the potential of immunocytochemical p16 and Ki-67 dual staining as compared to cytology, alone or in combination with HPV partial genotyping for triage in a cohort of 1763 HPV positive women attending a cervical cancer screening program. As compared to cytology, they found dual staining safer for the decision-making regarding HPV positive women’s need for follow-up with colposcopy and biopsy.

The study is well conducted and the paper well written. This is a significant contribution that should help to reduce the burden on colposcopy clinics that results from the introduction of the less specific HPV primary screening.

A few minor remarks

-       At the left bottom of the figure 1, the addition of the different histological diagnosis is 1320. It is not clear where does this figure come. Can you specify?

-       The gain in sensitivity of dual staining is counterbalanced by a lower specificity. This could be mentioned in the abstract without modifying its positive conclusion. For instance, by adding a short sentence at the end of the result paragraph.

-       It is stated in the discussion that “the explanation for the majority of 9% invalid p16/Ki-67 dual staining results was low cellularity and expected, as the smear for dual staining was reprocessed after the smear for cytology screening”. Since dual staining is a more complex technique than cytology, it is likely that a higher rate of analyses would be non-conclusive for technical reasons. Could the authors shortly mention what would be the workflow if dual staining was systematic, and in case of non-conclusive result. Would it still be possible to perform classical cytology ?

-       Title: Programme or Program ? 

Reviewer 2 Report

The study design and the study data are convincing and they result in scientifically valid diagnostic performance data.

The comparative interpretion of screening algorithms should however be more explanatory, since the concluding statements are not sufficiently clearcut as detailed below:

Regarding statement „testing with dual staining alone or in combination with partial genotyping by Cobas4800 is a safe alternative for triage of HPV positive women from an HPV screened population.”, HPV screening and this type of partial genotyping are identical steps at several screening algorithms over the world. So dual staining can hardly be a lone testing method, as positive HPV data should be present at each case to be tested by this advanced staining method.

Regarding statement „would provide a safer decision making for whether . . . or . . .”,  two choices are mentioned while the possiblities for two different HPV results by partial genotyping and two different staining results (i.e yes/no) results in four strata. So which strata would indicate colposcopy and which ones would indicate cytologic follow-up?

Regarding statement „replacing the current cytology screening with . . . would result in . . . an increased number of . . . CIN2+”, one would like to know, what would have been the fate these cytology undiagnosed CIN2+ lesions. Would they have regressed? Would they have been diagnosed at the next screening time still at intraepithelial stage? The authors investigated if further testing (such as dual staining) can refine deicision making and the results indicate that single point sampling will moderately fit to a refined decision making on biopsy. Instead, authors should speculate on that refined decision making would rather mean refined follow-up intervals for testing with less invasive methods.
